# Host-Induced Gene Silencing of an Adenylate Kinase Gene Involved in Fungal Energy Metabolism Improves Plant Resistance to *Verticillium dahliae*

**DOI:** 10.3390/biom10010127

**Published:** 2020-01-12

**Authors:** Xiaofeng Su, Guoqing Lu, Xiaokang Li, Latifur Rehman, Wende Liu, Guoqing Sun, Huiming Guo, Guoliang Wang, Hongmei Cheng

**Affiliations:** 1Biotechnology Research Institute, Chinese Academy of Agricultural Sciences, Beijing 100081, China; suxiaofeng@caas.cn (X.S.); luguoqing007@163.com (G.L.); lixiaokang2016@163.com (X.L.); latif_ibge@yahoo.com (L.R.); sunguoqing02@caas.cn (G.S.); guohuiming@caas.cn (H.G.); 2Department of Biotechnology, University of Swabi, Khyber Pakhtunkhwa 23561, Pakistan; 3State Key Laboratory for Biology of Plant Diseases and Insect Pests, Institute of Plant Protection, Chinese Academy of Agricultural Sciences, Beijing 100193, China; wendeliu@126.com; 4Department of Plant Pathology, Ohio State University, Columbus, OH 43210, USA

**Keywords:** host-induced gene silencing (HIGS), *Verticillium dahliae*, pathogenicity factor, adenylate kinase

## Abstract

Verticillium wilt, caused by the ascomycete fungus *Verticillium dahliae* (*Vd*), is a devastating disease of numerous plant species. However, the pathogenicity/virulence-related genes in this fungus, which may be potential targets for improving plant resistance, remain poorly elucidated. For the study of these genes in *Vd*, we used a well-established host-induced gene silencing (HIGS) approach and identified 16 candidate genes, including a putative adenylate kinase gene (*VdAK*). Transiently *VdAK*-silenced plants developed milder wilt symptoms than control plants did. *VdAK*-knockout mutants were more sensitive to abiotic stresses and had reduced germination and virulence on host plants. Transgenic *Nicotiana benthamiana* and *Arabidopsis thaliana* plants that overexpressed *VdAK* dsRNAs had improved *Vd* resistance than the wild-type. RT-qPCR results showed that *VdAK* was also crucial for energy metabolism. Importantly, in an analysis of total small RNAs from *Vd* strains isolated from the transgenic plants, a small interfering RNA (siRNA) targeting *VdAK* was identified in transgenic *N. benthamiana*. Our results demonstrate that HIGS is a promising strategy for efficiently screening pathogenicity/virulence-related genes of *Vd* and that *VdAK* is a potential target to control this fungus.

## 1. Introduction

*Verticillium dahliae* (*Vd*), the causal agent of Verticillium wilt (Vw), is a destructive fungal pathogen infecting over 400 plant species, including important ornamental, horticultural, agronomical, and woody plants [1]. Symptoms are not uniform among different plant species [2], and in general, the fungus is difficult to control because it survives in the plant vascular system and infects the hosts via infested soil and diseased plant debris [3]. Once the plants are infected, no fungicides can cure the plants [4]. *Vd* can cause severe economic losses worldwide. In China, approximately 250–310 million US dollar losses have been reported for cotton annually due to *Vd* [5]. Considerable studies on Vw and the associated fungi and the public release of the genomic sequence of *Vd* and its sister fungus *V. alfalfae*, have enabled the identification of candidate pathogenicity and virulence genes [6,7,8]. However, the complexity of the pathogenic mechanism is a serious constraint on developing effective management strategies. In addition, we still have little knowledge about virulence factors in *Verticillium* species [9,10,11,12], although a few genes from *Vd* are known to be involved in its virulence of this fungus. For instance, the deletion of the β-1,6-endoglucanase gene substantially reduced disease symptoms in eggplant by compromising the utilization of cellulose [13]. The adhesion protein Vta2 and transmembrane protein mucin Msb, which serve as positive regulators, enable microsclerotial formation, adhesive capacity, invasive growth, and/or sporulation [10,14]. VdMcm1, a MADS-box transcription factor, has multiple biological functions during spore and microsclerotial production, secondary metabolism, and virulence [15]. As a member of the Myosin V family, VdMyo5 may participate in vesicle transport to regulate vegetative growth and is essential for full virulence [16]. Although multiple genes playing a role in signal pathways, development, and nutrition have been reported [17,18,19,20], few have been confirmed as feasible candidates for the control of *Vd* in plants.

A host-induced gene silencing (HIGS) method for *in planta* RNAi silencing of a parasitic fungal gene achieved a breakthrough as a pathogen-derived resistance strategy to control fungi and other microorganisms [21,22,23,24,25]. With this method, manipulation of the accumulation of dsRNA that targets fungal transcripts in barley (*Hordeum vulgare*) and wheat (*Triticum aestivum*) inhibited the development of the powdery mildew fungus *Blumeria graminis* [26]. HIGS, guided by *Barley stripe mosaic virus* (BSMV), interfered with the expression of three predicted pathogenicity genes, thus suppressing invasion and colonization by the wheat leaf rust fungus, *Puccinia triticina* [27]. Synthetic dsRNAs could compromise expression of the targeted fungal gene and inhibited conidial germination of *Fusarium oxysporum* f. sp. *cubense* and *Mycosphaerella fijiensis* [28]. In a transgenic banana that expressed a hairpin RNA against *velvet* and *Fusarium transcription factor 1* genes of *F. oxysporum* f. sp. *cubense* (Foc), resistance was improved for 8 months, indicating potential use for breeding [29]. HIGS-assisted silencing of β-1,3-glucan synthase gene *FcGls1* caused elevated resistance against Fusarium head blight [30]. In recent independent studies, the HIGS strategy was deployed to interfere with desirable target genes of *Vd*, and new virulence factors, e.g., *RGS1*, *Ave1*, *Sge1,* and *NLP1*, were identified [31,32]. For instance, HIGS-assisted interference with *Vd RGS1* conferred elevated resistance in cotton [31].

In previous studies, we generated transgenic plants by overexpressing dsRNA against *VdAAC* (ADP, ATP carrier: Responsible for transferring ATP from the mitochondria into the cytoplasm) and *VdSTT3* (oligosaccharyl transferase subunit, playing an essential role in glycoprotein modification) to improve resistance against *Vd* [33,34]. To explore more candidate genes from *Vd* to target and further verify the use of the HIGS system, here we (1) screened for *Vd* candidate pathogenicity/virulence factors to take advantage of genomics data and the well-established HIGS approach; (2) characterized adenylate kinase (VdAK) functions in virulence; (3) evaluated the resistance level of transgenic *Nicotiana benthamiana*, and *Arabidopsis thaliana* harboring dsVdAK against *Vd* and (4) demonstrated that small interfering RNAs (siRNAs) can enter fungi from within the plants and improve the resistance of host plants.

## 2. Materials and Methods

### 2.1. Plant Materials and Growth Conditions

*N. benthamiana* plants were cultivated in pots containing sterilized soil (peat compost: vermiculite, 1:1, *w*/*w*) in a greenhouse (25 ± 2 °C and 75% relative humidity, 16 h light and 8 h dark). *A. thaliana* ecotype Columbia-0 was sown and grown in the growth chamber (23 ± 2 °C, 70% relative humidity, 16 h light and 8 h dark, light intensity 4000 lx).

### 2.2. Construction of HIGS and Transforming Plasmids and Plant Transformation

Total RNA was isolated using an RNA extraction kit (YPHBio, Tianjin, China), and 1 μg of the total RNA was used to synthesize the first-strand cDNA using a reverse transcription kit (TransGen, Beijing, China) according to the manufacturer’s instructions. The construction of HIGS plasmids was initiated using PCR amplification of target fragments of 400 to 600 bp. These sequence-verified target fragments were cloned into tobacco rattle virus (TRV) vector via double-digestion with BamHI and EcoRI and ligation with T4 ligase using standard protocols. TRV:00 or its derivatives were independently transformed into *Agrobacterium tumefaciens* strain GV3101 by electroporation [35]. Subsequently, the constructs were agroinfiltrated into seedlings with a needleless syringe [36,37]. The primers used for PCR amplification are listed in Appendix A.

Gateway cloning technology was used to construct the plasmid for expressing dsRNA of *VdAK*. The target fragment was produced by PCR amplification from *Vd* cDNA, then cloned into the intermediate vector pDONR207. The expression cassette pK7GWIWG2(I)::*VdAK* was obtained via attL and attR (LR) recombination reaction between the entry plasmid pDONR207::*VdAK* and the destination vector pK7GWIWG2(I) using Gateway LR Clonase II Enzyme Mix (Invitrogen, San Diego, CA). The sequence-verified plasmid was transformed into *A. tumefaciens* strain LBA4404 by electroporation to transform *N. benthamiana* and *A. thaliana* plants using a leaf disc co-cultivation method and floral dip method, respectively [38,39,40]. The primers are listed in Appendix A.

### 2.3. Assessment of Disease Resistance

The highly virulent wild-type *Vd* strain V991 was shake-cultured in a complete medium (CM) containing 50 mg/L ampicillin and 50 mg/L kanamycin at 25 °C for 6 days. The roots of seedlings were incubated for 2 min in a suspension of 1 × 10^6^ spores/mL, and immediately replanted into new pots. Disease severity (DS) at 10 to 12 days post-inoculation (dpi) was scored using previously described methods [33,41] as 0, no wilt; 1, less than 2 leaves wilting; 2, 3–5 leaves wilting; 3, more than 5 leaves wilting or chlorotic; and 4, plant death or near death, to calculate a disease index (DI): DI = [∑ (No. of plants × DS score)/(Total no. of plants × Highest DS score)] ×100.

The fungal biomass in tissues was quantified using a slight modification of a previously described method [12]. In brief, DNA was extracted using the Plant Genomic DNA Kit (TIANGEN, Beijing, China) separately from infected roots, stems (0–3 cm above the ground surface), and leaves of 5 plants. ITS1 and ITS2 regions of the ribosomal RNA genes (Z29511) of *Vd* were amplified to quantify fungal DNA as previously described [12].

Hyphal development in roots of host plants after inoculation with *Vd-GFP* (green fluorescent protein) and Δ*VdAK-GFP* strains (described next) was assessed at 5 dpi using a confocal microscope (Zeiss LSM 700, Jena, Germany).

### 2.4. VdAK Gene Disruption, Complementation, and VdAK-GFP Mutant Strains

The *VdAK* gene knockout plasmid was generated using a previously described method [33]. Briefly, ~1.2-kb upstream and downstream flanking fragments of *VdAK* and hygromycin resistance (*HPT*) gene expression cassette were fused by overlap PCR. The knockout fragment was obtained using nested PCR by fusion PCR product as the template, then cloned into the pGKO2 vector by attB and attP (BP) recombination reaction using Gateway BP Clonase II Enzyme Mix (Invitrogen, San Diego, CA, USA).

For the construction of the Δ*VdAK-GFP* strains, the neomycin resistance (Neo^R^) gene expression cassette amplified from pCAM-neo with TrpC promoter and TrpC terminator was inserted into the expression plasmid pCAMBIA1302 via XbaI and BstEII restriction sites. The *GFP* expression cassette was cloned into the plasmid via the XbaI and KpnI restriction sites to generate pCAMBIA1302::neo::GFP. Then, the GFP open reading frame (ORF) was replaced with the *VdAK* ORF via the ScaI and PstI restriction enzyme sites to generate pCAMBIA1302::neo::VdAK for Δ*VdAK-C*. 

Transformants, including the Δ*VdAK*, Δ*VdAK-C* strains and Δ*VdAK-GFP*, were obtained by protoplast transformation [42]. Positive transformants were selected by RT-PCR and parallel antibiotic resistance. The primers used in these constructions are listed in Appendix A.

### 2.5. Stress Treatments

A 10 µL drop of 1 × 10^6^ spores/mL of the respective strains was placed on a plate of Czapek-Dox agar amended with 0.5 M NaCl or 0.5 M sorbitol or treated with a 10 s pulse of 302 nm UV light [43]. Colony diameters were measured after 4 weeks. For estimating spore production, 3 mL of sterilized water was added to each plate, which was gently shaken to release the spores [12]. The spores were counted using a hemacytometer and light microscope (OLYMPUS BX52, Tokyo, Japan).

### 2.6. RT-qPCR

The expression of target genes was quantified by RT-qPCR using a 7500 Real Time PCR System (Applied Biosystems, Foster City, CA, USA) and TransStart Top Green qPCR SuperMix (TransGen, Beijing, China) according to the manufacturer’s instructions. The 2^-∆∆Ct^ method was used for relative quantification of transcripts for each gene [44]; the *N. benthamiana* housekeeping gene (*Nbactin*), *A. thaliana* housekeeping gene (*AtEF-1α*) or *Vd* housekeeping gene (*Vdactin*) was used as an internal control. The primers are listed in Appendix A.

### 2.7. Small RNA (sRNA) Sequencing and Data Analysis

The stems of *Vd*-infected seedlings were surface-disinfested and placed on potato dextrose agar (PDA). After 1 week, any fungal colony that had grown from the infected stems was cultured on PDA for observation and in CM broth for RNA extraction. sRNA library construction and sequencing were performed by Novogene (http://www.novogene.com/). Total sRNAs were mapped to the published genomes of *Verticillium* (Broad Institute, https://www.broadinstitute.org/). The targeting relationship between reads and targeted sequences of *VdAK* was predicted by miRanda (http://www.microrna.org/microrna/home.do). Information on sRNAs from *Vd* isolated from *Vd*-infected seedlings is listed in Appendix A.

### 2.8. Statistical Analysis

Data from 3 independent experiments were analyzed using Duncan’s multiple range test using SPSS Statistics 17.0 software (SPSS, Chicago, IL, USA).

## 3. Results

### 3.1. HIGS Candidate Pathogenicity Factors Were Selected Based on Available Protein-Encoding Genes

To identify the pathogenicity factor genes that are required for *Vd* virulence, we employed a well-established virus-guided HIGS system [45]. For the classification and annotation of the 10,535 publicly available predicted protein-encoding genes of the *Vd* strain VdLs.17 [6], GO analysis was carried out (Appendix A). Based on SwissProt and Blast results, these genes were divided into cellular components, molecular functions, and biological processes; 92 fungal genes involved in diverse biological processes (energy, metabolism, development, secreted protein, and others) were considered as candidates for HIGS in *N. benthamiana* system (Appendix A). They had no sequence similarity in an analysis of sequence similarity between these genes and the *N. benthamiana* transcriptome using online databases (https://www.ncbi.nlm.nih.gov/).

### 3.2. HIGS-Assisted Screening for the Candidate Genes

Ten days after infiltration, the seedlings were dipped into 10^6^ spores/mL of resuspended *Vd* conidia, and the DI was calculated from severity scores at 10–12 dpi (Appendix A). Plants that were injected with the TRV:00 (control group) showed typical wilting, stunting, chlorosis, and necrosis, and the DI reached 100 at 10 dpi. The plantlets in the RNAi groups, however, displayed varying levels of increased resistance to *Vd* compared with the control group. 

The DI for the different RNAi groups gradually increased over time from 10 dpi. According to the Chinese national criteria for evaluating tolerance to Vw in cotton (GB/T 22101.5-2009), a plant variety is tolerant to Vw when DI <35. Within the RNAi groups, the DI for infected plants infiltrated with any of the 16 RNAi constructs used in the HIGS was obviously reduced (Appendix A and Figure 1). These 16 candidates are involved in energy metabolism, material transportation, protein modification, glucose metabolism, cell proliferation, DNA replication, and resistance. These results suggested that HIGS of *Vd* genes could be deployed to identify candidate virulence factors.

### 3.3. VdAK May Be Responsible for Adaptation to Abiotic Stress

Based on the phenotypes and DI above, seedlings infiltrated with TRV:*VdAK* (VDAG_01040.1) displayed slight wilt. *AK* is a regulator of the metabolic pool of ATP, ADP, and AMP [46]. ADP and AMP are not only the source of energy [47], but also crucial factors in signal pathways for growth, development, and stress responses [48]. To ascertain the functions of *VdAK*, we generated disruption mutants (Δ*VdAK*) via homologous recombination, a GFP strain (Δ*VdAK*-*GFP*), and a complementation strain (Δ*VdAK-C*) via random insertion (Appendix A). The transformants were selected after three generations and confirmed by RT-PCR (Appendix A).

Subsequently, we investigated the putative roles of VdAK in adaptive responses of the fungus to various stresses. Strain Δ*VdAK* displayed no distinct defect in development or spore production, but growth and sporulation were significantly diminished by 0.5 M NaCl, 0.5 M sorbitol, and the 10 s UV treatment (Figure 2). Especially on media with NaCl and sorbitol, Δ*VdAK* colonies had statistically delayed hyphal growth and produced fewer spores than the wild-type *Vd* and Δ*VdAK*-*C* strains, which did not differ from each other. These results suggested *VdAK* has a positive role in fungal response to abiotic stress.

### 3.4. VdAK Is Required for Full Vd Virulence

We further investigated the virulence of Δ*VdAK*, Δ*VdAK*-*C,* and wild-type *Vd* strains by inoculating 6-week-old seedlings of *N. benthamiana* with respective conidia. At 12 dpi, wilting symptoms and leaf necrosis were not visible on plants inoculated with Δ*VdAK* mutants, but symptoms were visible on plants inoculated with wild-type *Vd* (Figure 3A). The biomass of strain Δ*VdAK* was much lower in roots, stems, and leaves relative to that of the wild-type *Vd* (Figure 3B). Complementation strain Δ*VdAK-C* did not differ in virulence or biomass from the wild-type *Vd* (Figure 3A,B). Furthermore, hyphae of both *Vd-GFP* and Δ*VdAK*-*GFP* had colonized the roots by 5 dpi. Notably, spore germination was lower, and hyphae less abundant for the strain Δ*VdAK*-*GFP* than for *Vd-GFP* on the root surface (Figure 3C). In *A. thaliana*, strain Δ*VdAK* was less virulent and produced less biomass than did the wild-type and complementary strains, whereas the virulence of Δ*VdAK-C* and the wild-type *Vd* were equivalent (Appendix A). Thus, these results confirmed that *VdAK* is indispensable for the full virulence of *Vd*.

### 3.5. dsVdAK-Overexpressing Transgenic Plants Had Significant Resistance against Vd

To further validate whether *VdAK* is associated with disease development, we generated stable transgenic lines of *N. benthamiana* and *A. thaliana* by inserting an RNAi construct with a 536 bp fragment of *VdAK* and the constitutive cauliflower mosaic viral 35S promoter. When 6-week-old T_2_ plants of three independent transgenic lines of *N. benthamiana* were tested for *Vd* resistance using the wild-type *Vd*, wilt symptoms were visibly milder at 12 dpi than on the control lines (Figure 4A). RT-qPCR analysis showed that the accumulation of fungal DNA was significantly suppressed in roots, stems, and leaves of the transgenic lines, relative to the wild-type (Figure 4B). The transcript level of *VdAK* in the transgenic lines was significantly reduced up to 4-fold compared with the wild-type (Figure 4C). In *A. thaliana*, the reduction of *VdAK* expression in transgenic plants resulted in increased *Vd* resistance and a reduction of fungal biomass (Appendix A). Thus, the results demonstrated that the knockdown of *VdAK* expression compromised the virulence of *Vd*.

### 3.6. VdAK Is Associated with Energy Metabolism

Among various proteins involved in energy metabolism, adenylate kinases have vital roles in maintaining an ADP/ATP balance [49,50]. In addition, the vacuolar ATPase (VA) can enhance vacuolar H^+^ pumping activity and Na^+^ compartmentalization capacity [51]. ATP6 is one of the three main subunits in membrane-localized ATP synthase F0, which is highly similar, functionally and mechanistically, to V-ATPase [52,53]. Furthermore, adenylate cyclase (AC), catalyzing the conversion of ATP to 3′,5′-cyclic AMP (cAMP), has key regulatory roles in signaling pathways [54]. ATP-phosphoribosyltransferase (ATP-PRT), the rate-limiting enzyme of the histidine pathway, is completely reversible depending on the ATP concentration [55,56]. To ascertain whether energy metabolism in *Vd* is affected with the knockout of *VdAK*, we quantified transcripts of the marker genes *VdVA*, *VdATP6*, *VdAC* and *VdATP-PRT* using RT-qPCR and harvested mycelium of strains Δ*VdAK*, Δ*VdAK*-*C*, and wild-type *Vd*, which had been cultured in CM broth for 7 days (Figure 5A–D). Transcript levels for *VdVA*, *VdATP6*, and *VdAC* were significantly elevated up to 2-fold, and *VdATP-PRT* transcripts increased significantly (>10-fold) in Δ*VdAK* when compared to levels in the Δ*VdAK*-*C* and wild-type *Vd*. Thus, manipulating the *VdAK* transcripts resulted in transcriptional reprogramming of these genes associated with energy metabolism.

### 3.7. A siRNA Targeting VdAK May Contribute to Cross-Kingdom Gene Silencing and Virulence Inhibition

To further corroborate the reduction in wilt, the HIGS experiment was repeated in *VdAK1*-transgenic lines (Figure 6). At 12 dpi, the transgenic seedlings appeared to have significant resistance compared with WT *N. benthamiana* seedlings. Colonies recovered from transgenic seedling stems also differed distinctly from those from the WT seedlings (Figure 6A). To investigate whether siRNAs were generated in transgenic seedlings and entered *Vd*, the isolated fungi were cultured in CM broth, and the total RNA of the mycelium was subsequently extracted and sequenced (Figure 6B). Among small RNAs, one was found to target VdAK specifically (Figure 6B; Appendix A). To validate HIGS, we further examined the *VdAK* transcripts in *Vd* using RT-qPCR. Transcript levels for *VdAK* were downregulated about 4-fold by HIGS (Figure 6C). Moreover, the expression of two marker genes, determined using RT-qPCR, was upregulated after *VdAK* silencing (Figure 6D,E). These results support the view that exogenous siRNAs can enter pathogenic fungi through plants and silence the target gene to improve host resistance.

## 4. Discussion

In this study, we first developed a HIGS system for large-scale screening of pathogenicity/virulence factors, and found 16 potential candidates. Subsequently, we confirmed that the VdAK plays a vital role in fungal metabolism, conidiation, and pathogenicity and could be a valuable gene to transform plants for increased resistance to *Vd*. Indeed, the RNAi-*VdAK* transgenic plants exhibited significant resistance to *Vd*. Notably, siRNA that targeted *VdAK* was identified in transgenic plants. It may prove to be useful for cross-kingdom gene silencing.

In eukaryotic organisms, RNA interference (RNAi) is a highly conserved mechanism, and a valid tool to knock down gene expression [57,58,59]. Based on this mechanism, the TRV-mediated gene silencing has been used to downregulate endogenous genes in plants [36,38,60]. Recently, HIGS has also proved to be a promising strategy for silencing a foreign gene in a host to investigate numerous plant–pathogen systems [61,62], including *Vd* in various hosts [33,34,63]. In the present study, positive plants at 10 days after infiltration showed distinct photobleaching, suggesting the massive presence of dsRNA and its active role in gene silencing [64,65,66]. In our study, seedlings that were inoculated with *Vd* developed varying degrees of wilting from 10 to 12 dpi. The whole process took about 25 days. Hence, HIGS can be utilized as an efficient platform for genome-wide high-throughput RNAi screening to identify pathogenic genes. We also screened for *Vd* candidate pathogenicity/virulence factor genes using publicly available genomic resources and the TRV system. Within a comparatively short time, we identified 16 candidate genes, that when silenced, resulted in significant resistance to *Vd*.

We used a GO analysis for our preliminary selection of candidate genes of *Vd* involved in highly conserved biological function. A similar approach has also been used for *Blumeria graminis* in which targeted silencing of important genes such as heat shock protein 70, 40S ribosomal protein, and ADP/ATP carrier protein led to reduced sporulation [26]. We focused on an in-depth analysis of one candidate gene, *VdAK*, that encodes a putative adenylate kinase, whose reduced expression led to an obvious decrease in the virulence of *Vd*. Adenylate kinase is a critical phosphotransferase that catalyzes the interconversion of ADP and ATP [67,68,69]. Transphosphorylation of adenine nucleotides regulates the cellular concentrations of ADP, ATP, and AMP, which directly affects the adenylate distribution during glucose and nucleic acid metabolism [70,71]. This gene is indispensable for controlling the balance between ATP and ADP in cells [67,72] and thus is important for adaptation to diverse stresses [73], as shown by our analysis of Δ*VdAK* strains exposed to various stresses. Under stress, strain Δ*VdAK* grew significantly less and produced fewer spores compared to the wild-type. This decrease was rescued by complementation with a functional *VdAK* gene. Using the *VdAK* disruption and *VdAK* complementation mutant strains, we found that disruption of *VdAK* increased fungal sensitivity to different stresses. Thus, VdAK apparently contributes to improved responses to diverse stresses.

Transcripts of *VdAC*, *VdATP6*, *VdAC,* and *VdATP-PRT* were upregulated in strain Δ*VdAK*, presumably triggered by a high ratio of ADP/AMP [74]. The upregulation of the vacuolar ATPase (*VA*) gene is associated with improved salt tolerance in halotolerant peppermint Keyuan-1 [51]. The upregulation of *VdVA*, *VdATP6*, and *VdAC* might also be related to ATP synthesis and the signal network [75,76,77]. When *VdATP-PRT* expression is strongly enhanced, glycometabolism may be stimulated and thus provide cellular energy [78,79,80]. Taken together, the accumulative evidence suggests that *VdAK* is a positive virulence regulator that is linked to energy metabolism. Moreover, the ability to germinate and colonize the root surface of *N. benthamiana* and *A. thaliana* may also be impaired in Δ*VdAK*, leading to less colonization in the plant vessels. Our results clearly indicate that the *VdAK* gene positively regulates virulence of *Vd*. Given the roles of AK in energy metabolism, when *VdAK* is suppressed, ADP/ATP turnover might also be disturbed, reducing fungal growth, development, and virulence. The compromised virulence and stress tolerance of the Δ*VdAK* disruption mutant support this hypothesis.

As a group of small, noncoding RNAs, siRNAs regulate post-transcriptional gene expression and participate in diverse biological processes, including resistance against stress [81,82,83,84]. In screening and study of a class of miRNAs related to *Phytophthora sojae* resistance in three soybean cultivars [85], the expression of miR393 in soybean significantly increases in response to *P. sojae* infection [86]. After *Vd* infection, cotton and *A. thaliana* increase the production of miRNA166 and miRNA159 to target *Vd* genes and result in improved resistance [87]. Five miRNAs in a highly resistant strain of *Vitis davidii*, revealed by microRNA sequencing, were specifically expressed and used to investigate further the potential inhibition of grape white rot disease caused by *Coniella diplodiwlla* [88]. In our study, a specific siRNA against *VdAK* was detected in *Vd* isolated from transgenic plants, which provides further insights into the action between siRNA and fungi *in planta*. *VdAK* was silenced due to the presence of siRNA, which results in the increased expression of *VdVA* and *VdATP6* involved in energy metabolism. These results are consistent with the expression of the marker genes in Δ*VdAK* strains. The reduced wilt symptoms are likely caused by the disruption of energy metabolism of *Vd* and subsequent growth in transgenic seedlings. These results illustrate that the *VdAK* gene has potential as another target for a HIGS strategy to control *Vd*.

## 5. Conclusions

In summary, we confirmed that the HIGS system is very efficient for screening candidate pathogenic factors in *Vd*. As a positive regulator needed for full virulence, VdAK holds promise as a target to enhance the resistance of transgenic plants harboring dsVdAK against *Vd*.

## Figures and Tables

**Figure 1 biomolecules-10-00127-f001:**
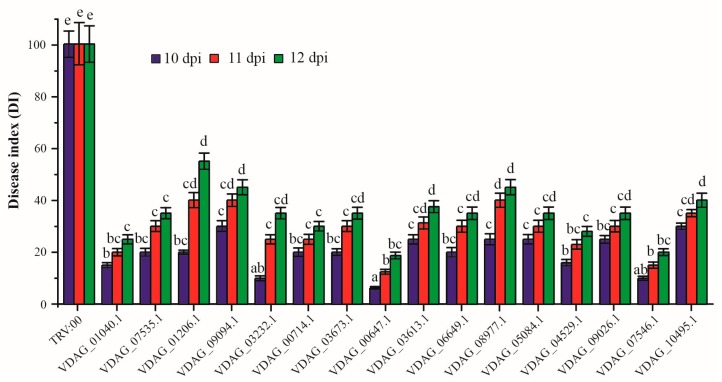
Disease index over time for *N. benthamiana* seedlings infiltrated with 16 *Vd* strains that had been silenced for a candidate virulence-related gene.

**Figure 2 biomolecules-10-00127-f002:**
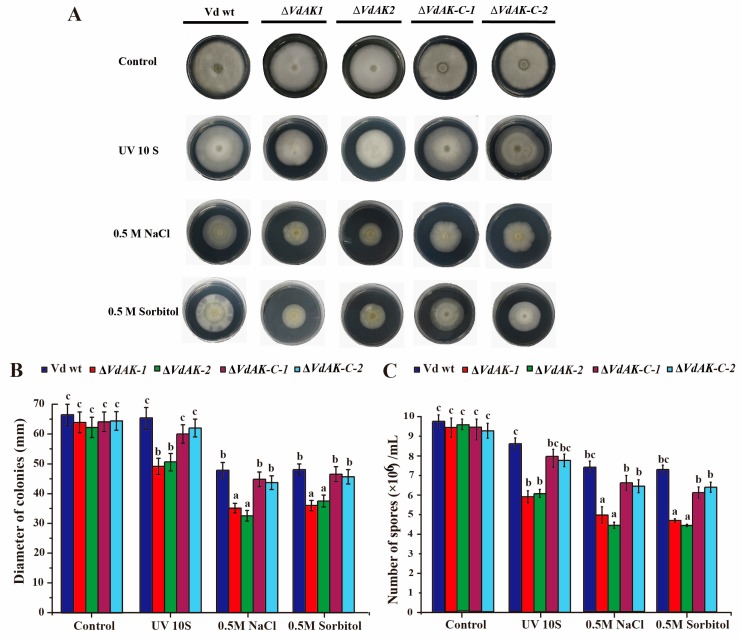
The deletion of *VdAK* compromises mycelial growth and spore production during abiotic stress. A 10 µL drop of 10^6^ spores/mL was placed in the center of Czapek Dox agar in a petri dish. Colony morphology (**A**), colony diameter (**B**), and spore production (**C**) of disruption (Δ*VdAK*) and complementation (Δ*VdAK-C*) mutants and wild-type *Vd* strain exposed to three abiotic stresses were analyzed after 4 weeks. Means (±SE) from three independent experiments were analyzed for significant differences among treatments using Duncan’s test (*p* <0.05). Different letters above bars within a treatment indicate significant differences among the strains.

**Figure 3 biomolecules-10-00127-f003:**
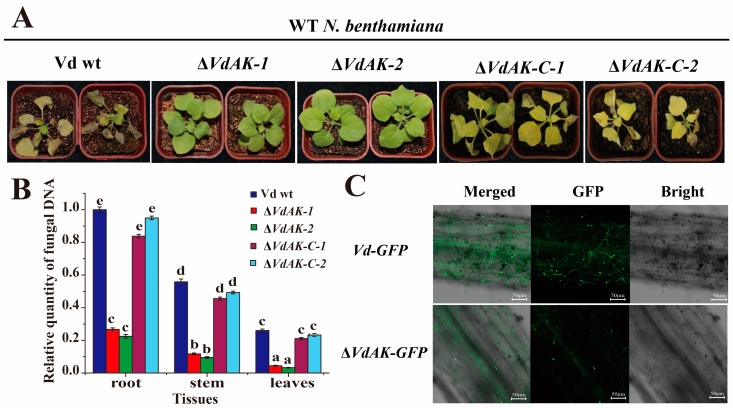
Virulence analysis of disruption (Δ*VdAK*), complementation (Δ*VdAK-C*), and wild-type *Vd* strains in *N. benthamiana*. (**A**) Symptoms on seedlings 12 days after roots were dipped in 10^6^ spores/mL of Δ*VdAK*, Δ*VdAK-C,* or wild-type *Vd* (Vd wt). (**B**) Relative amounts of fungal DNA as determined by RT-qPCR. Means (± SE) from three independent experiments were analyzed for significant differences among treatments using Duncan’s test (*p* <0.05), as indicated by different letters. (**C**) Micrographs of fluorescing hyphae in *N. benthamiana* root tips at 5 dpi with *Vd-GFP* or Δ*VdAK-GFP*.

**Figure 4 biomolecules-10-00127-f004:**
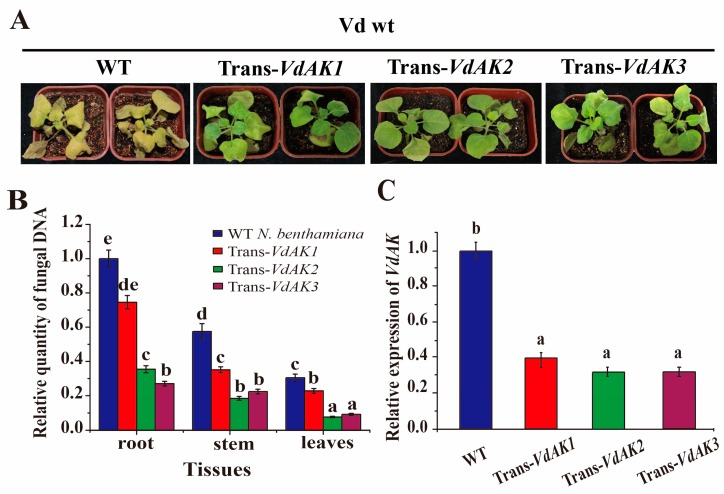
Expression of dsRNA of *VdAK* improved *Vd* resistance in *N. benthamiana* compared to wild-type plants (WT). WT and trans-*VdAK* plants were inoculated with wild-type *Vd* (Vd wt). (**A**) Symptoms on transgenic and wild-type plants at 12 days after roots were dipped in 10^6^ spores/mL of the respective strains. (**B**) Relative amounts of fungal DNA as determined by RT-qPCR. (**C**) Transcript levels of *VdAK* in stems of transgenic and WT plants. Means (± SE) from three independent experiments that differed significantly among treatments in Duncan’s test (*p* <0.05) are indicated by different letters.

**Figure 5 biomolecules-10-00127-f005:**
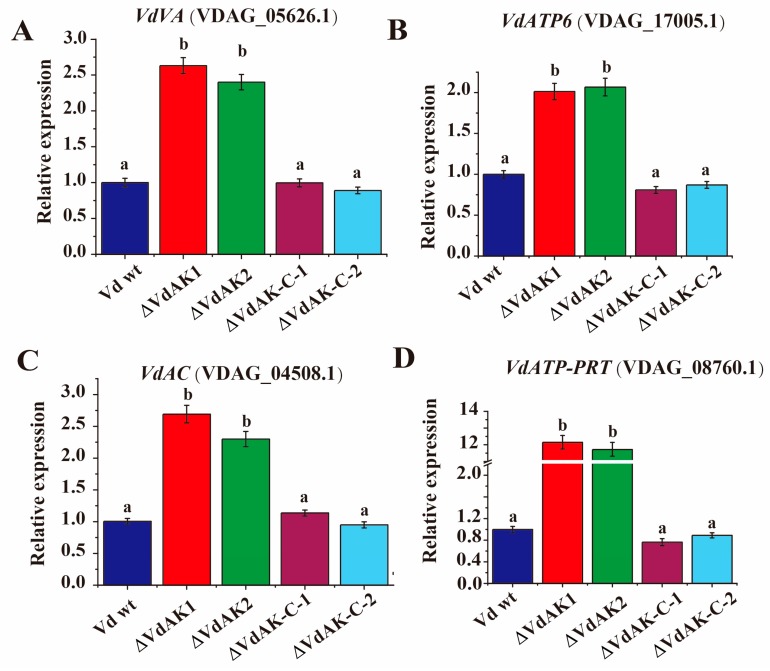
Relative expression of fungal genes associated with energy metabolism in Δ*VdAK*, Δ*VdAK-C,* and wild-type *Vd* (Vd wt) strains. Strains were cultured in complete medium (CM) broth for 7 days. Transcript levels of *VdVA* (**A**), *VdATP6* (**B**), *VdAC* (**C**), and *VdATP-PRT* (**D**). Mean (± SE) levels from three independent experiments that differed significantly in Duncan’s test (*p* <0.05) are indicated by different letters.

**Figure 6 biomolecules-10-00127-f006:**
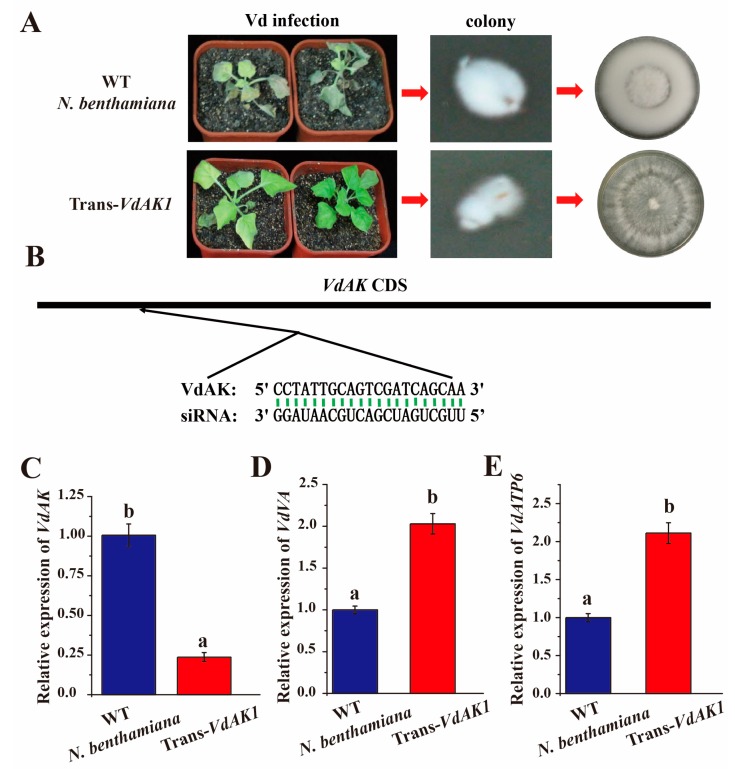
Host-induced gene silencing of *VdAK* enhanced resistance against *V. dahliae* (*Vd*). (**A**) Disease symptoms on *N. benthamiana* at 12 dpi and growth phenotype on PDA of *Vd* isolated from infected seedlings. (**B**) Specific siRNA targeted the *VdAK* position. The siRNA sequence aligned with *VdAK* at the predicted binding site. Relative expression levels of *VdAK* (**C**), *VdVA* (**D**), and *VdATP6* (**E**) in RT-qPCR analysis of the recovered mycelium. Mean (± SE) from three independent experiments that differed significantly in Duncan’s test (*p* <0.05) are indicated by different letters.

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
