# Peer review of "Host-Induced Gene Silencing of an Adenylate Kinase Gene Involved in Fungal Energy Metabolism Improves Plant Resistance to Verticillium dahliae"

_biomolecules, 2020, doi:10.3390/biom10010127_

Round 1

Reviewer 1 Report

Line 23: dozens? Please specify exact amount

Line 23: Factors? Greater specificity needed.

Figure 1: error bars needed on DI – standard error or standard deviation would be appropriate.

Line 197: VdAK may be responsible…

Line 207: index - > indices

Figure 2: colony diameters? I think figure S3 should be included in the main document.

Line 217: Checked -> Investigated

Line 227: showed that -> confirm

Author Response

Dear Reviewer,

Thank you on behalf of all the authors for your review of “Host-induced gene silencing of an adenylate kinase gene involved in fungal energy metabolism improves plant resistance to Verticillium dahliae” (biomolecules-676687). We responded to all the suggested revisions point-by-point as below.

Answers to your suggestions:

According to your positive comments, we carefully revised the manuscript and did our best to improve it. Extensive English editing was completed by Beth E. Hazen (PhD), a professional in scientific editing and writing. Meanwhile, it was further polished by Dr. Hayat Khan, Department of Microbiology, The University of Swabi, Pakistan. Furthermore, we have checked the whole manuscript for any mistake.

Line 23: dozens? Please specify exact amount

Answer: As per the instruction, dozens have been replace with the exact amount. Please see line 25.

Line 23: Factors? Greater specificity needed.

Answer: We have changed it. Please see line 26.

Figure 1: error bars needed on DI – standard error or standard deviation would be appropriate.

Answer: It is a good idea. We have added the standard error for the data in Figure 1.

Line 197: VdAK may be responsible…

Answer: As recommended, the title has been changed. Please see Line 212.

Line 207: index - > indices

Answer: As per your suggestion, we have changed the expression style. Please see line 222.

Figure 2: colony diameters? I think figure S3 should be included in the main document.

Answer: Fig. 2 and Fig. S3 have been merged. Please see Fig. 2.

Line 217: Checked -> Investigated

Answer: The word “checked” has been replaced with “investigated”. Please see Line 240.

8.Line 227: showed that -> confirm

Answer: Agreed. We have changed it. Please see Line 251.

Reviewer 2 Report

The manuscript by Su et al. "Host-induced gene silencing of an adenylate kinase gene involved in fungal energy metabolism improves plant resistance to Verticillium dahliae" is generally well written and scientifically sound.  The results are interesting and will spur further work in this area.  I have no major issues or concerns but would like the following more minor issues to be addressed:

Many grammatical errors throughout, as limited examples:  line 71 – compromised should be compromise. Line 325, sentence must have a determiner – A similar approach, not Similar approach (this type of error is found throughout and makes the manuscript difficult to follow).

Additional issues to address:

Line 112 – table S3 mentioned before table S2.  Please correct/reorder. 

It is not clear to me how you selected your target genes in section 3.2 from the list of 92 potential genes identified in section 3.1?  What was your criteria for selecting this subset of 16 from the list of 92?

Likewise, what was the rationale in selecting VDAG_01040.1 in section 3.3 over the remaining 15 other genes.  For example, it appears VDAG_00647.1 had the greatest effect on reducing DI – why was this gene not selected?

Line 207 – index is singular, this index was or these indices were

Line 227 – you state VdAK is dispensable for virulence.  I think you mean to say indispensable?  As you have it written, you are saying VdAK is not needed for virulence.  That does not seem to be the case based on your data.

First sentence, line 324-325 does not make sense.  Please clarify this sentence.

Author Response

Dear Reviewer,

It is in response to your suggestions regarding our manuscript entitled “Host-induced gene silencing of an adenylate kinase gene involved in fungal energy metabolism improves plant resistance to Verticillium dahliae” (biomolecules-676687). We would like to thank you on behalf of all the authors for reviewing our manuscript. We have responded to all the suggested revisions point-by-point as below.

Responses to your comments:

Many grammatical errors throughout, as limited examples: line 71-compromised should be compromise. Line 325, sentence must have a determiner-A similar approach, not Similar approach.

Answer: According to your positive comments, we have carefully revised the manuscript for any grammatical errors and corrected the mistakes. Pleasure see line 77 and 359-360. We did our best to improve the manuscript as suggested. Meanwhile, the manuscript was further polished by Beth E. Hazen (PhD), a professional in scientific editing and writing, and Dr. Hayat Khan, Department of Microbiology, The University of Swabi, Pakistan.

The other revisions are listed as follows.

Line 112: table S3 mentioned before table S2. Please correct/reorder.

Answer: As per the recommendations, supplementary materials have been reordered throughout the manuscript. Please see line 120, 153, 167, 176, 202 and 439-443.

It is not clear to me how you selected your target genes in section 3.2 from the list of 92 potential genes identified in section 3.1? What was your criteria for selecting this subset of 16 from the list of 92?

Answer: There is a Chinese national criterion (GB/T 22101.5-2009) for the selection of tolerant plants against Verticillium wilt, according to which, a variety is considered tolerant to Vw if the DI (disease index) is less than 35. At 10 dpi, the DI of 16 RNAi groups was less than 35 and the candidate genes were selected for further investigation. Please see line 199-202 for the addition of this criterion in the manuscript.

Likewise, what was the rationale in selecting VDAG_01040.1 in section 3.3 over the remaining 15 other genes. For example, it appears VDAG_00647.1 had the greatest effect on reducing DI-why was this gene not selected?

Answer: According to the reference, VDAG_01040.1 is essential gene involved in energy metabolism and is highly conserved in many physiological races of Verticillium. Thus, we firstly selected it to confirm the HIGS experiment. The biological function of remaining genes are now further investigated in our lab.

Line 207: index is singular, this index was or these indices were

Answer: As per the instruction, we have changed the expression style. Please see line 222.

Line 227: you state VdAK is dispensable for virulence. I think you mean to say indispensable? As you have it written, you are saying VdAK is not needed for virulence. That does not seem to be the case based on your data.

Answer: Thank you for pointing it out, yes we mean indispensable but due to typing error it was written as dispensable. Now we have corrected it to “indispensable”. Please see line 251.

First sentence, line 324-325 does not make sense. Please clarify this sentence.

Answer: The sentence has been rephrased, kindly see line 357-359.